# Reliable Cell Trackers Say "I dunno!"

## Abstract

Cell tracking is a key computational task in live-cell microscopy, but fully automated analysis of high-throughput imaging requires reliable and, thus, uncertainty-aware data analysis tools, as the amount of data recorded within a single experiment exceeds what humans are able to overlook. We here propose and benchmark various methods to reason about and quantify uncertainty in linear assignment-based cell tracking algorithms. Our methods take inspiration from statistics and machine learning, leveraging two perspectives on the cell tracking problem explored throughout this work: Considering it as a Bayesian inference problem and as a classification problem. Our methods admit a framework-like character in that they equip any frame-to-frame tracking method with uncertainty quantification. We demonstrate this by applying it to various existing tracking algorithms including the recently presented Transformer-based trackers. We demonstrate empirically that our methods yield useful and well-calibrated tracking uncertainties.

## 1 Introduction

Uncertainty-aware cell tracking is a key requirement for fully automated data analysis of high throughput live-cell microscopy (LCM) data, where images often contain hundreds or thousands of almost indistinguishably looking, moving, growing and dividing cells and where temporal resolution of the time-lapses is limited by biological and technical considerations such as phototoxicity (Tinevez et al., 2012) or camera movement speed in multi-colony setups (Seiffarth et al., 2025). LCM enables researchers to analyze cellular behavior beyond the population level, revealing dynamics, development and multi-species interactions (Blöbaum et al., 2024; Fante et al., 2024; Burmeister et al., 2021). The recent advances in computer vision, mainly driven by the success of deep learning and ever-improving computational resources, form a substantial pillar of modern LCM analyses, as the amount of data collected in a single experiment exceeds what humans are able to overlook (Kasahara et al., 2023; Seiffarth et al., 2025; Witting et al., 2025). The strong reliance on computational tools demands for high reliability and trustworthiness, which is improved by uncertainty-aware analyses aiming to reliably estimate the confidence in their own predictions. While over-confidence in deep neural networks encountering distribution shifts is a commonly known issue in the deep learning community (Guo et al., 2017; Kristiadi et al., 2020), uncertainty estimation as a remedy has – so far – attracted only little attention in cell tracking.

In this work, we strive to make cell tracking more reliable by complementing it with uncertainty estimation. To this end, we explore two perspectives on the tracking problem, considering it as a Bayesian inference and as a classification problem. Our Bayesian perspective gives rise to the *cell tracking posterior*, motivating sampling and approximate sampling methods for uncertainty quantification, which however come at increased computational costs. The classification perspective provides less costly, nevertheless useful alternatives to quantify uncertainty. Moreover, the classification perspective also provides tools to evaluate and calibrate uncertainty estimates using known

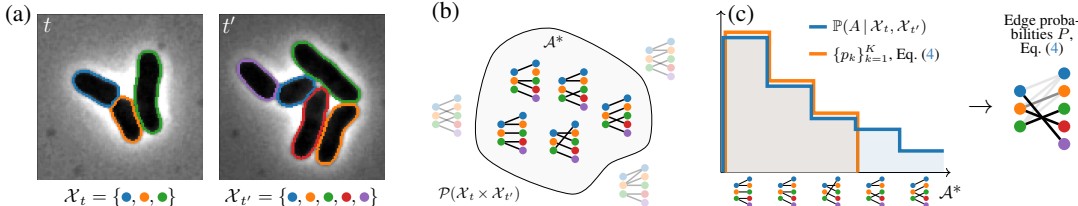

Figure 1: Schematic presentation of our Bayesian perspective and inference method for uncertainty-aware cell tracking. a depicts the detections $\mathcal{X}_t, \mathcal{X}_{t'}$ of two consecutive frames. b The set of biologically feasible assignments $\mathcal{A}^*$ forms a subset within the set of all possible many-to-many assignments. c Sorting assignment solutions by their posterior density we approximate the full posterior by a set of the $K$ most plausible solutions.

techniques such as *temperature scaling* (Guo et al., 2017). Notably, the methods under consideration are applicable to the large family of linear assignment-based tracking methods, which are embedded naturally within our framework.

## 2   Probabilistic Cell Tracking

**Cell Tracking as Bayesian Inference**   In the case of multiple cells $x_1, \ldots, x_m \in \mathcal{X}_t$ and $x'_1, \ldots, x'_n \in \mathcal{X}_{t'}$, we aim to find an *assignment* $A \in \mathcal{P}(\mathcal{X}_t \times \mathcal{X}_{t'})$, such that every mother cell is represented at most twice and every daughter cell is represented at most once. We denote the subset of all assignments that adhere to those constraints as $\mathcal{A}^* \subset \mathcal{P}(\mathcal{X}_t \times \mathcal{X}_{t'})$, the *biologically feasible assignments* (*cf.* Figure 1b). The common approach to solve for a single assignment is to optimize the joint likelihood of observing cells $\mathcal{X}_{t'}$ given $\mathcal{X}_t$ and an assignment $A$, *i.e.*

$$A_{\text{opt}} := \underset{A \in \mathcal{A}^*}{\arg\max}\ \mathbb{P}(\mathcal{X}_{t'} \mid \mathcal{X}_t, A) = \underset{A \in \mathcal{A}^*}{\arg\max} \sum_{(x_i, x'_j) \in A} -w(x_i, x'_j) - m w_a - n w_d \tag{1}$$

where $m$ and $n$ are the numbers of appearing and disappearing cells, $-w_a$ and $-w_d$ are the respective log probabilities of the events, and $w(x_i, x'_j)$ is some chosen cost function. Considering that the structure of the solution $A \in \mathcal{A}^*$, *i.e.* its biological feasibility, is known *a priori* without the need for any actual observations, the formulation from Equation (1) lends itself nicely to a Bayesian interpretation with $A_{\text{opt}}$ being the *maximum a posteriori* (MAP) estimate of the posterior distribution of assignments

$$\mathbb{P}(A \mid \mathcal{X}_{t'}, \mathcal{X}_t) \propto \mathbb{P}(\mathcal{X}_{t'} \mid \mathcal{X}_t, A)\, \mathbb{P}(A) \tag{2}$$

where we choose the uniform distribution over valid assignments $\mathcal{A}^*$ as prior distribution, $\mathbb{P}(A) := \mathbb{1}_{\{A \in \mathcal{A}^*\}} / |\mathcal{A}^*|$. Given this *cell tracking posterior* distribution, we obtain the predictive distribution of the event that $x_i$ is the mother of $x'_j$

$$P_{ij} := \mathbb{P}((x_i, x'_j) \in A \mid \mathcal{X}_t, \mathcal{X}_{t'}) = \sum_{A \in \mathcal{A}^*} \mathbb{1}_{\{(x_i, x'_j) \in A\}}\, \mathbb{P}(A \mid \mathcal{X}_t, \mathcal{X}_{t'}) \tag{3}$$

as the weighted frequency of $x_i$ being the mother of $x'_j$ among all possible solutions. We estimate Equation (3) by means of the self-normalized importance-weighted estimator (Tokdar and Kass, 2010)

$$P_{ij} \approx \sum_{k=1}^{K} \mathbb{1}_{\{(x_i, x'_j) \in A_k\}} \cdot p_k, \quad \text{with weight} \quad p_k := \frac{\mathbb{P}(\mathcal{X}_{t'} \mid \mathcal{X}_t, A_k)\, \mathbb{P}(A_k)}{\sum_{l=1}^{K} \mathbb{P}(\mathcal{X}_{t'} \mid \mathcal{X}_t, A_l)\, \mathbb{P}(A_l)}, \tag{4}$$

for $k = 1, \ldots, K$ and where we now assumed $A_1, \ldots, A_K$ to be the top-$K$ best solutions. We sketch this method in Figure 1c and refer to it as AS ▬ in the results. As an alternative approach, we consider perturbation of of input features like the detected masks or positions. We propose two approaches: FP ▬, which averages the costs over the perturbed features, and FP+A ▬, which solves the assignment problem for each pair of perturbed input features and approximates the predictive posterior Eq. (3) as the frequency at which an event $\{(x_i, x'_j) \in A\}$ is contained in the resulting assignment solutions.

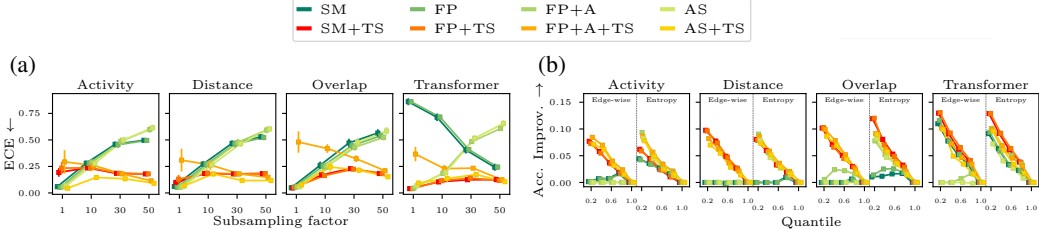

Figure 2: (a) Mean ECE and standard error thereof across datasets at decreasing temporal resolution. (b) Mean accuracy improvement when only considering the most certain tracking predictions, using either the edge-wise confidence (*cf.* Eq. (5)) or entropy (*cf.* Eq. (6)) as uncertainty estimate.

**Cell Tracking as Classification Task**   Cell tracking can be viewed as a multi-class classification problem by considering the problem of choosing the right mother for each individual daughter. That is, for your favorite daughter $x'_j \in \mathcal{X}_{t'}$, the *classes* are just the mothers $\mathcal{X}_t$ from the preceding frame. The class – or rather mother – probabilities are obtained by performing daughter-wise softmax normalization of the predicted costs

$$P_{i|j} = \mathbb{P}((x_i, x'_j) \in A \mid x'_j, \mathcal{X}_t, \mathcal{X}_{t'}) = \frac{\exp\{-w(x_i, x'_j)\}}{\sum_{x \in \mathcal{X}_t} \exp\{-w(x, x'_j)\}}. \tag{5}$$

We can interpret the softmax probabilities $P_{i|j}$ from Equation (5) as confidence scores, enabling their usage for uncertainty estimation. This approach is also compatible to the Bayesian approaches from Section 2 by simply column-normalizing the edge probabilities $P_{ij}$ from Equation (4) as $P_{i|j} = P_{ij}/\sum_k P_{kj}$. Moreover, the classification view enables to estimate and improve calibration using temperature scaling (TS ▬) as in Guo et al. (2017). The latter however is only applicable if annotated tracking data is available. Finally, as an alternative measure of uncertainty we can also consider the *daughter-wise entropy* $\mathcal{H}_j$ of the conditional distribution $P_{i|j}$

$$\mathcal{H}_j = -\sum_i P_{i|j} \log P_{i|j}. \tag{6}$$

# 3   Results

In our experiments we investigate the calibration of tracking algorithms by Ruzaeva et al. (2022, Activity), Crocker and Grier (1996, Distance), Fukai and Kawaguchi (2023, Overlap) & Gallusser and Weigert (2025, Transformer). We consider 2D+t datasets from the Cell Tracking Challenge (Maška et al., 2014, 2023) and a large-scale microbial time-lapse microscopy sequence (Seiffarth et al., 2025). Our results show that these tracking algorithms are not *per se* well-calibrated (*cf.* SM ▬, FP ▬, FP+A ▬ & AS ▬ in Fig. 2a), but calibration is achievable using TS, if annotated tracking data is available (*cf.* SM+TS ▬, FP+TS ▬, FP+A+TS ▬ & AS+TS ▬ in Fig. 2a). Further, we consider the accuracy improvement achievable by sorting out the most uncertain tracking decisions and present results in Fig. 2b. This setting reflects a possible real-world scenario where a human annotator is guided by the uncertainty estimate to manually correct the automated tracking result. We notice that using our proposed daughter-wise entropy approach for quantifying uncertainty, even the vanilla tracking algorithms can produce 'useful' uncertainties, *i.e.* uncertainties that positively correlate with model performance. However, in particular our approximate Bayesian methods FP+A ▬ & AS ▬ show higher improvements in accuracy and are less reliant on temperature scaling and, thus, annotated tracking data. This comes at the price of increased computational costs of those methods. Overall, we also observe that using the entropy-based uncertainty estimation (*cf.* Eq. (6)) is more reliable in this setting.

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
