# OpenReview forum: "Reliable Cell Trackers Say "I dunno!""
_EurIPS.cc/2025/Workshop/MedEurIPS — EurIPS 2025 Workshop MedEurIPS Submission_

### Official Review · Reviewer_MRVB · 2025-10-31
**Interesting cell tracking work**

**Rating:** 9
**Confidence:** 4

**Review:**

This work is about automated cell tracking with uncertainty qualification. The work is very close to medical domain. It is also very interesting work as they see the question from 2 perspective, the Bayesian inference way, and the classification way. I believe this work will bring a lot interesting discussion.

---

### Official Review · Reviewer_q8Y3 · 2025-10-31
**Review comments**

**Rating:** 7
**Confidence:** 5

**Review:**

This paper presents a  unified framework with cell tracking bayesian inference and classification to quantify cell tracking uncertainty. This framework is broadly applicable to the large family of linear assignment based tracking methods. The paper is well written and easy to follow, with novel contribution to live-cell microscopy image analysis and broader medical image tracking applications.

---

### Decision · Program_Chairs · 2025-10-31

**Decision:**

Accept (Oral)

**Comment:**

Both reviewers highlight the paper’s novelty and clarity. The proposed Bayesian and classification-based framework for quantifying tracking uncertainty is well motivated and relevant to medical image analysis.